# ROBUST FORECASTING OF NETWORK SYSTEMS SUBJECT TO TOPOLOGY PERTURBATION

## ABSTRACT

Many real-world dynamical systems, such as epidemic, traffic, and logistics networks, consist of sparsely interacting components and thus naturally exhibit an underlying graph structure. Forecasting their evolution is computationally challenging due to high dimensionality and is further complicated by measurement noise and uncertainty in the network topology. We address this problem by studying the predictability of graph time series under random topology perturbations, a problem with major implications that has remained largely unexplored. In the limit of large networks, we uncover distinct noise regimes: systems that are predictable with arbitrary accuracy, systems predictable only up to limited accuracy, and systems that become entirely unpredictable. Motivated by this characterization, we propose a time series forecasting framework based on a probabilistic representation of network dynamics, which leverages Bayesian coreset approximations for scalable and robust dimentionality reduction. Numerical experiments on both synthetic and real-world networks demonstrate that our approach achieves competitive accuracy and robustness under topology uncertainty, while significantly reducing computational costs.

## 1 INTRODUCTION

Network structure is ubiquitous in real-world interconnected systems, making the study of robustness of network forecasting a topic of major importance. It spans applications ranging from epidemic spread prediction to logistics and traffic network forecasting, among many others. Yet, robustness studies have so far been restricted to static network tasks (Ni et al., 2024; Zügner and Günnemann, 2019; Wang et al., 2021; Li et al., 2024a; Wang et al., 2024), such as node classification, edge classification or graph regression... As for data-driven modeling of non-linear network dynamical systems, it has been mostly focused on proposing various time-varying graph neural network architectures (Liu and Zhang, 2024; Lan et al., 2022; Shao et al., 2022; Yan et al., 2024b) for settings with given noiseless topologies, besides signal recovery schemes (Sardellitti et al., 2021; Ceci and Barbarossa, 2018). Hence, we consider in this work the problem of forecasting network systems under mis-specification of the topology, in a high-dimensional setting i.e. for networks with a large number of nodes. Such types of noise or uncertainty may be encountered due to several reasons, including partial observability such as in social networks, abrupt changes such as road closures due to accidents, or equipment failure in electrical grids. Addressing this problem entails designing a suitable scheme that aims to circumvent the sensitivity to topology perturbation by probabilistic estimation of the main components of the considered state system evolution, i.e. a robust model reduction scheme. Specifically, the key contributions are organized as follows:

- We provide a theoretical analysis of the impact of topology perturbation on the system trajectory, for a class of common network systems.
- We develop a Bayesian coreset reduction of Graph Convolution Network (GCN) embeddings, resulting in a low-dimensional representation of network trajectories that is robust to random

topology perturbations. This representation naturally lends itself to Recurrent Neural Network (RNN)-based temporal modeling, giving rise to a forecasting scheme that is both robust and scalable.

- We conduct numerical experiments on simulated data of Kuramoto networks as well as real-world traffic data, demonstrating the competitiveness of the proposed approach.

The rest of the paper is organized as follows. In section 2, we review the related works and contrast them with the question we address. In section 3, we introduce the problem setting followed by the theoretical analysis of predictability under various noise regimes in section 4. In section 5, we present the proposed forecasting method followed by numerical result comparisons against the state-of-the-art in sections 6. Last, we conclude with a discussion of limitations and outlook in section 8.

## 2 RELATED WORKS

**Time-Varying Graph Neural Networks.**
In order to leverage the expressive power of graph neural networks (Kipf and Welling, 2017) for graph time series forecasting, several architectures have been proposed. First approaches were based on evolving the graph extracted features using recurrent neural networks (Manessi et al., 2020; Seo et al., 2018) or leveraging a RNN to evolve the weights of a graph convolution network (Pareja et al., 2020). More recently, various adaptive attention-based mechanisms for spatial or temporal modeling have been proposed (Guo et al., 2021; Yan et al., 2024a). For instance, Lan et al. (2022) combine a multi-order Chebyshev polynomial GCN with an adaptive self-attention mechanism to leverage the dynamic spatial correlation within multi-scale neighborhoods, whereas citepshao2022decoupled combines a self-attention layer with a GRU (Cho et al., 2014) to model the non-diffusive component in traffic forecasting. Several approaches leverage ODEs for the temporal modeling (Li et al., 2024b; Huang et al., 2020; Luo et al., 2023), offering a natural framework for handling irregularly sampled observations and learning latent continuous-time dynamics. We refer to Yan et al. (2024a) for an extensive review. Another recent direction targeting long-range forecasting is based on state-space models (Rahman and Coon). Yet, these approaches assume the graph to be given or lack scalability. In contrast, we propose a forecasting scheme that is designed to be robust to graph topology perturbation or misspecification.

**Robustness of Graph Neural Networks.** For static tasks such as node classification or edge classification, the robustness of GNNs has been extensively studied (Zügner and Günnemann, 2019; Wang et al., 2021; Bojchevski and Günnemann, 2019; Yang et al., 2024a;b; Geisler et al., 2021), mostly focusing on adversarial defense to node or edge attacks. Specifically, Entezari et al. (2020); Wu et al. (2019) propose pre-processing techniques to overcome adversarial perturbations via low-rank approximation of the graph adjacency matrix or gradient averaging, while Zhang and Zitnik (2020) propose a mechanism to tackle adversarial training by assigning higher weights to edges connecting similar nodes. On the other hand, Wang et al. (2021); Zügner and Günnemann (2019) propose certifiable defenses against bounded adversarial attacks, via convex relaxation of the robust optimization target or random smoothing. More recently, Yang et al. (2024a) proposed the first deterministic certificate defense leveraging a majority vote among sub-graphs defined via an unperturbed hash function. Nonetheless, most of these approaches are too conservative for non-adversarial noise settings and do not scale to time-varying graphs, given the exponential explosion in the number of sub-graphs to consider when a temporal component is introduced.

**Reduced Order Modeling.**
Model reduction of parametric evolution equations governing physical systems has been extensively studied (Benner et al., 2015), given the high computational cost associated with full-scale resolution. Specifically, two main families of methods have emerged: model-based projections and data-driven surrogates. Model-based approaches leveraging the structure of the equations include Galerkin projections (Hesthaven and Warburton,

2007), Krylov subspace methods (Liesen and Strakos, 2013), and dynamic low-rank approximations (Kazashi et al., 2025; Musharbash et al., 2020). They are based on different identification schemes of the underlying low-dimensional manifold capturing most of the variability of the system. More recently, several deep learning-based approaches have demonstrated competitive performance, among which Dynamic Mode Decomposition (Schmid, 2022), Physics-Informed neural networks (Cai et al., 2021) and Neural Operator Learning (Lu et al., 2021; Li et al., 2021) are most notable. Nonetheless, most work has been restricted to low-dimensional noiseless dynamical systems. Alternatively, we consider noisy high-dimensional network systems.

## 3 PROBLEM SETTING

A large class of natural and social processes, including those of epidemic spreading in populations, traffic flow in urban networks, synchronization in power grids, and gene regulation in biological systems, can be described as high-dimensional dynamical systems evolving over a network. Despite their diversity, the underlying dynamics often share a common structure: each node evolves according to its own self-dynamics while interacting with its neighbors through the network topology. Formally, this can be expressed as

$$\frac{dx_i(t)}{dt} = f(x_i(t)) + \sum_{j=1}^{n} a_{ij} g(x_i(t), x_j(t)), \ \ i = 1, \ldots, n \tag{1}$$

with node states $(x_1, \ldots, x_n)$, where $f$ describes the self-dynamics of $x_i$, $g$ captures the interactions with its neighbors, and $A = (a_{ij})_{1 \leq i,j \leq n}$ non-negative weights encoding the network topology. A large number of real-world dynamic networks can be modeled as such, including epidemic (Gao and Yan, 2022), traffic (Ding et al., 2019), and gene regulation (Aubin-Frankowski and Vert, 2020) networks, among many others (Prasse and Van Mieghem, 2022). Assume $f$ and $g$ to be unknown, but instead trajectories of the system nodes $(x_1(t_j), \ldots, x_n(t_j))_{j \leq m}$ are given, for different initial conditions. The task we address in this work is the forecasting of node states for unseen initial conditions as well as for $t > t_m$, as long as $(x_1(t), \ldots, x_n(t))$ is defined, under perturbation or misspecification of the network topology. We consider discrete random perturbations represented as a matrix $(\varepsilon_{i,j})_{1 \leq i,j \leq n}$ of i.i.d. Bernoulli random variables with success parameter $p \in (0, 1)$.

While equation 1 provides a general model for network dynamics, the critical question remains: *to what extent can such systems be predicted when the underlying topology is perturbed?* This is an open problem of both theoretical and practical importance, as robustness of forecasts directly depends on whether the dynamics remain predictable under noisy or uncertain topologies. In the next section, we establish distinct noise regimes that characterize when reliable forecasting is possible and when it inevitably breaks down.

## 4 FORECASTING SENSITIVITY TO NOISE

In this section, we investigate the predictability of network systems with noisy topologies. Specifically, in the limit of a large number of nodes, we identify distinct noise regimes. Surprisingly, under small noise, the system remains arbitrarily predictable. With weak noise, predictability persists but is limited in accuracy. As expected, higher levels of random perturbations render the system effectively unpredictable. More precisely, we present two results analyzing discrete and continuous perturbations respectively.

**Proposition 4.1.** *(Discrete Noise)*
*Consider a binary adjacency matrix $A = (a_{i,j})_{1 \leq i,j \leq n}$ and a discrete noise matrix $\varepsilon = (\varepsilon_{i,j})_{1 \leq i,j \leq n}$ of identically distributed Bernoulli random variables, with success probability $p \in (0, 1)$. Assume $f$ and $g$ to be continuously differentiable and the system trajectory to be supported on a space of lower dimension*

$m \ll n$, where $n$ is the number of nodes. Then, denoting by $y_1, \ldots, y_m$ the spatial modes and $(x_\varepsilon(t))_{t \in T}$ the trajectory over a compact time domain of the perturbed system

$$\frac{dx_i(t)}{dt} = f(x_i(t)) + \sum_{j=1}^{n} (a_{ij} + \varepsilon_{i,j}) \, g(x_i(t), x_j(t)), \quad i = 1, \ldots, n \qquad (2)$$

- If $p \leq \frac{(\max_k \|y_k\|_\infty)^{-1}}{n^{1+\alpha}}$ with $\alpha > 0$, then $\lim_{n \to +\infty} \mathbb{E}[\sup_{t \in T} \|x(t) - x_\varepsilon(t)\|] = 0$

- If $p = \frac{(\max_k \|y_k\|_\infty)^{-1}}{n}$, then there exists $M > 0$, such that for all $n \geq 2$,
$$\mathbb{E}[\sup_{t \in T} \|x(t) - x_\varepsilon(t)\|] \leq M$$

- If $p > \frac{(\max_k \|y_k\|_\infty)^{-1}}{n}$ and $\|g\|_\infty > \delta$, then for all $n > 2$, $\mathbb{E}[\sup_{t \in T} \|x(t) - x_\varepsilon(t)\|] > \delta$.

*Proof.* The proof is postponed to Appendix A.1. $\qquad\qquad\qquad\qquad\qquad\qquad\qquad\qquad\qquad$ □

**Proposition 4.2.** *(Gaussian Noise)*
*Consider a Gaussian noise matrix $\varepsilon = (\varepsilon_{i,j})_{1 \leq i,j \leq n}$ of identically distributed centered Gaussian random variables, with variance $\sigma^2 > 0$. Assume $f$ and $g$ are continuously differentiable and the system trajectory to be supported on a space of lower dimension $m \ll n$, where $n$ is the number of nodes. Then, denoting by $y_1, \ldots, y_m$ the spatial modes and $(x_\varepsilon(t))_{t \in T}$ the trajectory over a compact time domain of the perturbed system (2), we have*

- If $\sigma \leq \frac{(\max_k \|y_k\|_\infty)^{-1}}{n^{1+\alpha}}$ with $\alpha > 0$, then $\lim_{n \to +\infty} \mathbb{E}[\sup_{t \in T} \|x(t) - x_\varepsilon(t)\|] = 0$

- If $\sigma = \frac{(\max_k \|y_k\|_\infty)^{-1}}{n}$, then there exists $M > 0$, such that for all $n \geq 2$,
$$\mathbb{E}[\sup_{t \in T} \|x(t) - x_\varepsilon(t)\|] \leq M$$

- If $\sigma > \frac{(\max_k \|y_k\|_\infty)^{-1}}{n}$ and $\|g\|_\infty > \delta$, then for all $n > 2$, $\mathbb{E}[\sup_{t \in T} \|x(t) - x_\varepsilon(t)\|] > \delta$.

*Proof.* The proof is postponed to Appendix A.2. $\qquad\qquad\qquad\qquad\qquad\qquad\qquad\qquad\qquad$ □

**Remark.** (Approximability)
Given the invariance encoded in network systems, the set of their node trajectories typically lives (approximately) on a space of much lower dimensionality than the number of nodes. In particular, several common network models satisfy this property (Prasse and Van Mieghem, 2022).

**Remark.** (Practical Implications)
The previous results characterize precisely the fact that, for a large class of network systems, topology perturbation doesn't render predictability unachievable. This motivates the design of a network forecasting scheme that is least sensitive to network perturbation. We design such a scheme and demonstrate its practical performance in the following sections.

## 5 NETWORK CORESET FORECASTING

The theoretical analysis in Section 4 establishes that, under certain noise regimes, forecasting network dynamics remains feasible. Motivated by this characterization, we now design a forecasting scheme that explicitly aims to preserve predictability in the presence of topology perturbations. Our approach, termed Network Coreset Forecasting (NCF), leverages probabilistic reduction via Bayesian coresets to identify a

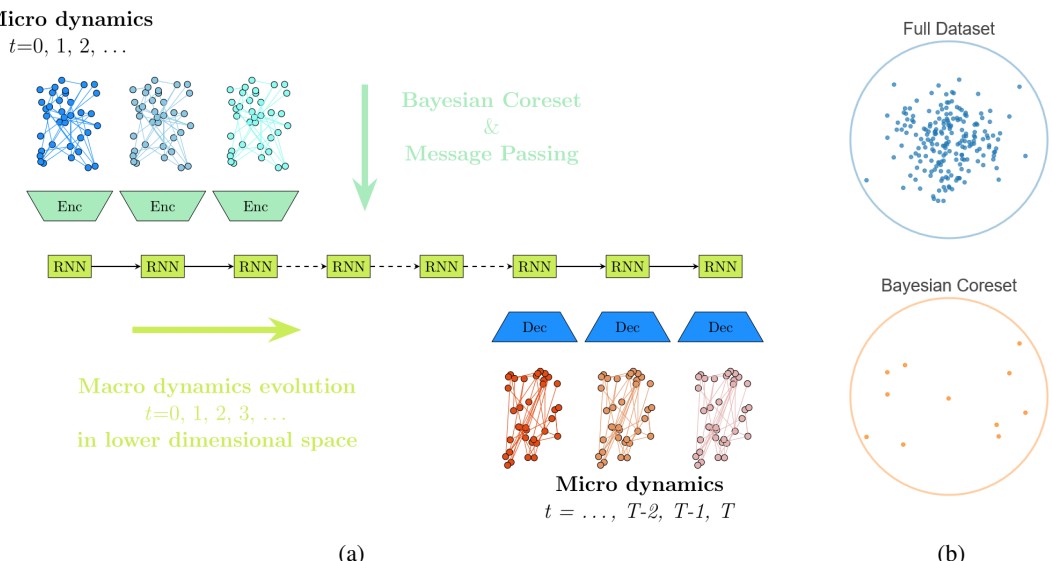

Figure 1: (a) Network Coreset Forecasting Scheme Illustration (b) Coreset Selection Illustration

compact set of representative node embeddings, which are then evolved in time for efficient and robust prediction. Hence, the key problem that we aim to solve is how to efficiently down-sample the processed node features for accurate prediction, while keeping sensitivity to network topology perturbation as small as possible. Classical approaches such as max-pooling (Hamilton et al., 2017) or low-rank approximations (Savas and Dhillon, 2011) either suffer from poor performance or high computational cost. Indeed, the nodes with the highest values at different screen-shots, might be very different from the ones that should be tracked to approximate the system trajectory. As for combinations of nodes with high eigenvalues, they can be quite sensitive to random perturbations, in addition to being costly to compute i.e. they require $O(n^3)$ operations. Consequently, we propose to leverage a probabilistic representation of the network time series by selecting the nodes which approximate the best, the distribution of the whole system trajectory. Specifically, we consider a Bayesian setting where the network time series constitute realizations of a given distribution and identify a subset of nodes $\{i_k, k \leq m\}$ such that the posterior given node embeddings $\{i_k, k \leq m\}$ is the closest to the full posterior. That is, we identify a Bayesian coreset (Campbell and Broderick, 2019; Huggins et al., 2016). In the following, we describe with greater detail the different components of the proposed method, illustrated in figure 1 and report a pseudo-code of the proposed algorithm in appendix D.

## 5.1 BAYESIAN CORESET APPROXIMATION

Bayesian coresets have been developed to reduce the cost of Bayesian inference with a large amount of data, without compromising accuracy (Campbell and Broderick, 2019). More precisely, considering a data set $(x_i)_{i \leq n}$ of $n$ observations, a likelihood $p(x_i|\theta)$ for each observation given the parameter $\theta \in \Theta \subseteq \mathbb{R}^d$, and a prior density $\pi_0$ on $\Theta$, the Bayesian posterior is given by

$$\pi(\theta) := \frac{1}{Z} \exp\left(\mathcal{L}(\theta)\right) \pi_0(\theta),$$

where the log-likelihood $\theta \mapsto \mathcal{L}(\theta)$ and the marginal likelihood $Z$ are defined by

$$\mathcal{L}(\theta) := \sum_{i=1}^{n} \mathcal{L}_i(\theta), \ \text{ s.t. } \ \mathcal{L}_i(\theta) := \log p(x_i|\theta) \ \text{ and } \ Z := \int \exp(\mathcal{L}(\theta))\pi_0(\theta)\mathrm{d}\theta.$$

The aim of the Bayesian coreset framework is then to find a set of weights $(w_i)_{i \leq n}$ such that

$$\min_{w \in \mathbb{R}^n} \left\| \mathcal{L} - \sum_{i=1}^{n} w_i \mathcal{L}_i \right\|_{\pi,\mathcal{L}} \ \text{ with } \ w \geq 0, \ \text{ and } \ \sum_{i=1}^{n} \mathbb{I}_{[w_i>0]} \leq m,$$

where $\| \cdot \|_{\pi,\mathcal{L}}$ is a functional norm that involves a scaling by the full likelihood $\mathcal{L}$ and the posterior $\pi$. This problem can be solved by a Frank-Wolfe type algorithm (Jaggi, 2013; Campbell and Broderick, 2019). We refer to Campbell and Broderick (2019) for more details. In our setting, we consider the (embedded) trajectory $(x_i(t_1), \ldots, x_i(t_\ell))$ of each node $i \leq n$ to be a multi-dimensional realization of a probability distribution, where $\ell$ is predicted sequence length, and we look for the subset of nodes that summarizes the best the whole network trajectory-segment. To simplify the algorithm and reduce the computational burden, we approximate the data distribution as Gaussian and adopt a Gaussian prior in the implementation.

**Remark.** (Computational Complexity)
The computational complexity of the Bayesian coreset reduction component may scale like $O(n^2)$ which is already an improvement from classical low-rank approximations which scales like $O(n^3)$. Hence, to improve scalability further, we use *random projections* for the computation of the norms, following Campbell and Broderick (2019). This results in a complexity of $O(nq)$, where $q$ is the dimension of the projection space.

## 5.2 GRAPH STRUCTURE EMBEDDING

Following previous work (Liu and Zhang, 2022; 2024), we extract node embeddings incorporating the graph information with a graph convolution network (Kipf and Welling, 2017). Specifically, we train a GCN encoder-decoder network, where the encoder performs message passing between nodes leveraging the network topology information and the decoder performs a diffusion of the updated states of the coreset selected nodes. For that matter, we first compute the coresets -kept nodes in reduced space, of the training data considering trajectories divided according to forecasting range target -forecasting sequence length, then we train the GCN decoder to estimate the updates of nodes discarded during the space reduction. Once this graph autoencoder is trained, we use its encoder to embed the node features before recomputing a new set of coresets that will be evolved in time by training a RNN, as we describe in the next subsection.

## 5.3 LATENT REPRESENTATION TEMPORAL EVOLUTION

Once the embedded node subsets are extracted, we simply train a long short-term memory (LSTM) network (Cho et al., 2014) to model the time evolution. As a result of the reduction in dimensionality, the temporal evolution computational cost is considerably reduced allowing for scalability to large networks. Note that such a reduction is necessary since the complexity of effective sequence-to-sequence models (e.g. RNNs, Transformers) is in $O(n^2)$, where $n$ is the dimension of the feature space. In the case of RNNs, that comes from the standard choice of a hidden-state dimension that scales similarly to input space dimension, resulting in an output scaling in $O(n^2)$. Once the predictions in reduced space are obtained, the node updates are diffused using the decoder. Given the probabilistic formulation of the model reduction component, the number of nodes which are kept slightly varies across trajectory segments. Hence, we consider the size of the largest coreset as RNN input dimension, and pad the input with the mean encoded value for smaller coresets.

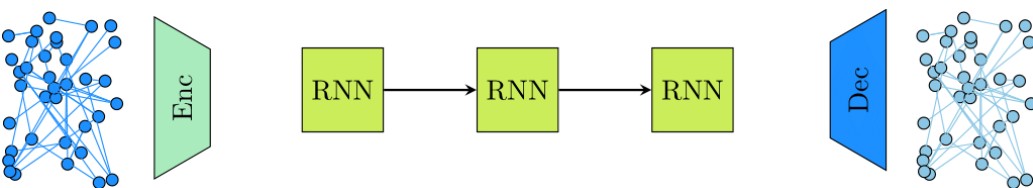

Figure 2: Time Evolution in Reduced Space

**Remark.** (Model Reduction Error Bound)
We note that a Bayesian coreset-based model reduction enjoys competitive error bounds in comparison to classical computationally efficient network reduction schemes. More precisely, we recall the bound derived by Campbell and Broderick (2019), for a given confidence level $1 - \alpha$, for $\alpha \in (0, 1)$ by

$$\left\| \mathcal{L} - \sum_{i=1}^{n} w_i \mathcal{L}_i \right\|_{\pi, \mathcal{L}} \leq \frac{\sigma \bar{\eta}}{\sqrt{m}} \left( 1 + \sqrt{2 \log \frac{1}{\alpha}} \right)$$

where

$$\bar{\eta} := \max_{i,j \in [N]} \left\| \frac{\mathcal{L}_i}{\sigma_i} - \frac{\mathcal{L}_j}{\sigma_j} \right\|_{\pi, \mathcal{L}}, \qquad \sigma_i := \| \mathcal{L}_i \|_{\pi, \mathcal{L}} \quad \text{and} \quad \sigma := \sum_{i=1}^{N} \sigma_i.$$

That is, the dimension of the reduced space to achieve a given accuracy level is chosen adaptively and depends on how well the trajectories align with one another. This is in contrast to selecting high-degree nodes, which might lead to arbitrary large error.

## 6 NUMERICAL RESULTS

We evaluate the proposed method on real-world traffic forecasting datasets, made publicly available by Guo et al. (2021). Traffic forecasting represents a major practical problem that requires robust forecasting schemes, given the different noise sources leading to topology changes that it can be subject to, such as weather incidents, traffic accidents as well as closure due to road maintenance. We compare our method against the state-of-the-art method $D^2STGNN$ proposed by Shao et al. (2022) and very recently shown to perform the best across datasets in the benchmarking study (Liu et al., 2023). Additionally, we compare against $ASTGNN$ which was proposed to handle heterogeneity of spatial-temporal graph data (Guo et al., 2021). Evaluation on test data is carried out using the Mean Absolute Error (MAE) in all displayed figures. We train the competitor methods with the hyper-parameters proposed by the respective authors, but truncate training to a computational time similar to that of NCF, for fair comparison. This is motivated by the goal of designing computationally efficient forecasting methods under limited budget, as discussed in previous sections. We explore the robustness of different methods with respect to perturbation by discrete i.i.d. noise -specifically Bernoulli matrices on challenging settings with high levels of noise. We report additional experiments, as well as an execution times comparison in Appendix B. We note from figures 3, 5 and 6 that NCF significantly outperforms the competitor methods. This is partly due to the highly noisy nature of the considered dataset, illustrated in figure 4, as is often the case in real-world settings. It's worth noting that NCF outperforms the state-of-the-art competitors even when there is no noise in the topology, which corresponds to noise level 0 in figures 3, 5 and 6. To further investigate robustness to structured changes in the topology as opposed to random ones, we estimate sparse approximations to the considered network topology for PEMS04, and explore the performance of the different methods under varying sparsity levels. We report the estimation procedure in Appendix C. We note that across models, datasets and perturbations above a certain noise level the performance, stabilizes. This is due to the fact that once the network information is destroyed, the

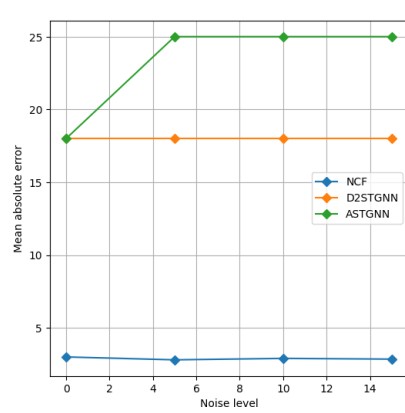

Figure 3: Test error evolution for PEMS03 dataset - 30% corrupted test data with discrete noise

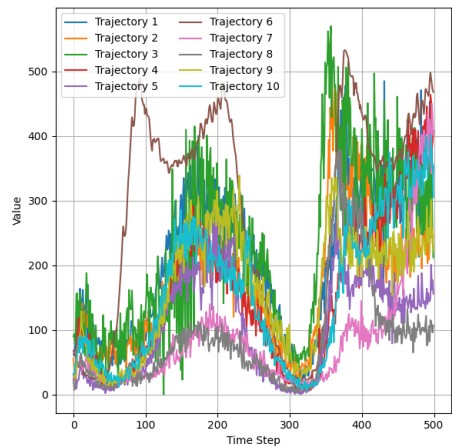

Figure 4: State trajectory evolution from the PEMS04 dataset

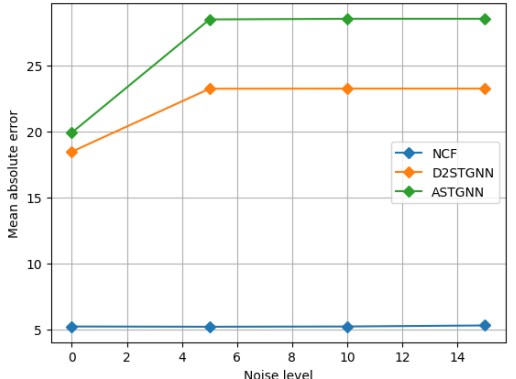

Figure 5: Test error evolution for PEMS04 dataset - 30% corrupted test data with discrete noise

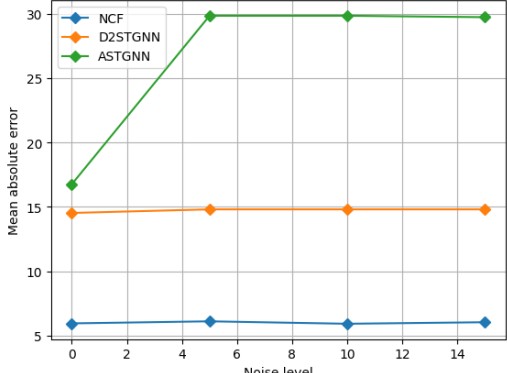

Figure 6: Test error evolution for PEMS08 dataset - 30% corrupted test data with discrete noise

forecasting schemes no longer exploit the topology information. We further evaluate NCF in correlated noise settings in appendix B.6. From a qualitative perspective, we analyze the adaptively selected nodes by NCF in comparison to nodes with high centrality, high variance or maximal value. We note that NCF selection closely tracks the global behavior of the system as expected unlike the other selection methods. We report in figure 8, the corresponding results for the social network model dynamic model proposed by Li et al. (2024b). We report further results in appendix B.7

## 7 LIMITATIONS & OUTLOOK

The proposed NCF approach performs competitively, but also raises limitations that point to opportunities for future research. First, it relies on a two-stage training approach as opposed to an end-to-end training formulation. Besides, since it relies on a probabilistic formulation of the model reduction component, it

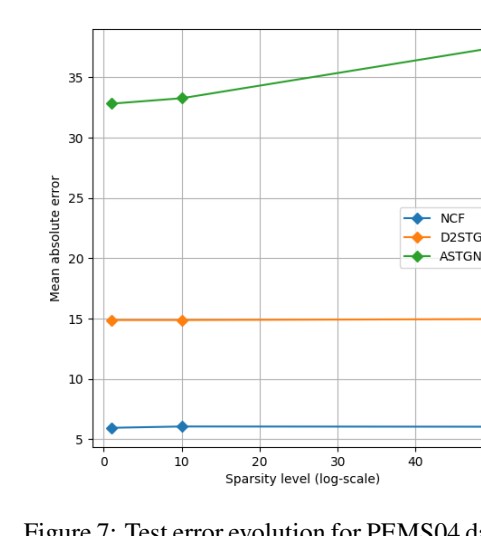

Figure 7: Test error evolution for PEMS04 dataset for varying sparsity levels

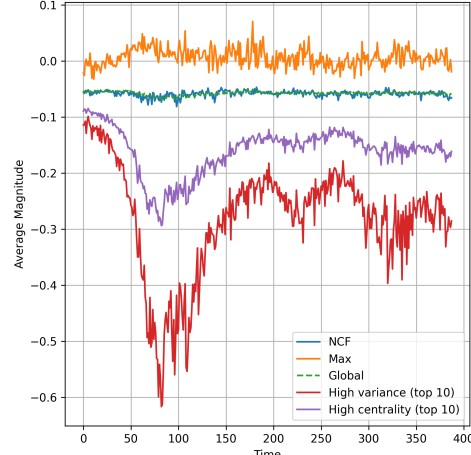

Figure 8: Average value evolution of selected nodes on a social web network model

might under-perform under perfect knowledge of the system trajectories, as compared to other approaches. We investigate this aspect by testing the different forecasting methods on a synthetic dataset based on the Kuramoto system of ODEs, which is used across the biological, chemical and electrical domains to simulate circadian oscillators, pacemaker cells in the heart and electrical power networks among other applications (Discacciati and Hesthaven, 2021; Dörfler and Bullo, 2014). For the topology, we generate a scale-free network instantiated as an Erdos-Renyi graph with connection probability between nodes equal to 0.3. We report in figures 10 the results for a network of size $n = 500$. We note from figure 10 that NCF under-performs for smooth uncertainty-free trajectories as those illustrated in figure 9. This comes from the fact that NCF relies on a probabilistic representation and hence introduces by design a level of non-smoothness through the randomized selection of the reduced model, although randomization is also what allows scalability. Consequently, when trained on clean data, like the smooth solutions of a Kuramoto system, it doesn't perform as well as randomization-free approaches like D2STGNN. This is in contrast with the performance obtained for real-world trajectories in the previous section. However, this is not a surprising aspect given that all learning methods are bound by the accuracy-robustness trade-off (Owhadi et al., 2015). Regarding sample complexity, since NCF is based on deep learning components, it requires a fairly high amount of data unlike state-space type models (Rahman and Coon). Despite that, NCF leads to significantly higher performance in high data-regime as we illustrate in appendix B.4. Overall, this constitutes a first study on robustness to topology perturbation and further evaluation of NCF on datasets such as METR-LA and PEMS-BAY (Shao et al., 2022) would be valuable to further assess its generalization, as well as to reduce the uncertainty observed for competitor models making the statistical significance stronger.

## 8 CONCLUSION

In this work, we addressed the open problem of forecasting network dynamical systems under uncertainty in their underlying topology. We characterized the sensitivity of network trajectories to random perturbations, identifying noise regimes that outline when reliable prediction is feasible and it when necessarily breaks down. On the algorithmic side, we developed a Bayesian coreset formulation of network forecasting, which yields a low-dimensional representation of network dynamics. Combined with GCN embeddings and RNN

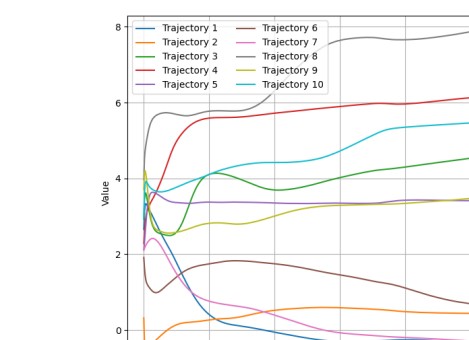

Figure 9: State trajectory evolution for the Kuramoto model - noise-free

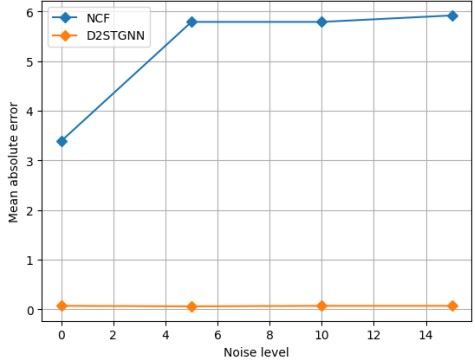

Figure 10: Test error evolution for Kuramoto network of size n=500 - 30% corrupted test data with discrete noise

temporal modeling, this approach provides a robust and scalable forecasting scheme. Our results integrate both theoretical analysis and probabilistic model reduction: the former clarifies the fundamental limits of predictability, while the latter delivers a practical and affordable method for achieving it. The evaluation on real-world datasets confirms that the proposed approach outperforms state-of-the-art baselines in noisy settings. By combining noise-regime analysis with a Bayesian coreset–based reduction, we provide a methodology that can be extended to more complex setting of structured/correlated perturbations with direct impact on domains such as epidemiology, transportation, and power systems where topology uncertainty is inherent.

**Reproducibility and LLM Usage.** We provide an implementation of the proposed method in Appendix E. The publicly available code by Guo et al. (2021) and Shao et al. (2022) is used for baselines. Parts of the manuscript text were refined with the assistance of a large language model. The technical content, derivations, and experiments are original to the authors.

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

# A PROOFS

## A.1 PROOF OF PROPOSITION 1

Recall that we would like to show that a Bernoulli perturbed network system features different behaviors depending the success parameter $p \in (0, 1)$ of the Bernoulli noise. Specifically,

$$p \leq \frac{(\max_k \|y_k\|_\infty)^{-1}}{n^{1+\alpha}} \text{ with } \alpha > 0 \quad \Rightarrow \quad \lim_{n \to +\infty} \mathbb{E}[\sup_{t \in T} \|x(t) - x_\varepsilon(t)\|] = 0$$

Denote by $Y$ the matrix of reduced space basis of the system trajectory, and $E$ the noise matrix i.e.

$$Y = \begin{pmatrix} y_1^\top \\ \vdots \\ y_m^\top \end{pmatrix}, \quad \text{and} \quad E = \begin{pmatrix} \varepsilon_1 \\ \vdots \\ \varepsilon_n \end{pmatrix}, \quad \text{with} \quad \varepsilon_i = (\varepsilon_{i,j})_{j \leq n}$$

Given a perturbed adjacency matrix $\hat{A} = A + E$, note that we have for $k \leq m$,

$$\|\hat{A}y_k - Ay_k\|_1 \leq \|Ay_k - Ay_k\|_1 + \|Ey_k\|_\infty = \|Ey_k\|_1$$

Furthermore, setting without loss of generality $x(1) = 0$ and $T = (1, 2)$, we have

$$\sup_{t \in T} \|x(t) - x_\varepsilon(t)\|_1 = \sup_{t \in T} \left\| \int_1^t \sum_{i,j} \varepsilon_{i,j} g(x_i(s), x_j(s)) \, ds \right\|_1$$

$$\leq \sup_{t \in T} \int_1^t \left\| \sum_{i,j} \varepsilon_{i,j} g(x_i(s), x_j(s)) \right\|_1 ds$$

$$\leq \int_1^2 \left\| \sum_{i,j} \varepsilon_{i,j} g(x_i(s), x_j(s)) \right\|_1 ds$$

leading to

$$\mathbb{E}\left[\sup_{t \in T} \|x(t) - x_\varepsilon(t)\|_1\right] \leq \mathbb{E}\left[\int_1^2 \left\| \sum_{i,j} \varepsilon_{i,j} g(x_i(s), x_j(s)) \right\|_1 ds\right]$$

By regularity of $g, x, x_\varepsilon$ and Fubini theorem, we get

$$\mathbb{E}\left[\sup_{t \in T} \|x(t) - x_\varepsilon(t)\|_1\right] \leq \int_1^2 \mathbb{E}\left\| \sum_{i,j} \varepsilon_{i,j} g(x_i(s), x_j(s)) \right\|_1 ds$$

Hence, by proposition 2 in section C of Prasse and Van Mieghem (2022), it is enough to show for each $k \leq m$ that

$$\mathbb{E}[\|Ey_k\|_1] \underset{n \to +\infty}{\longrightarrow} 0$$

That is equivalent to

$$\forall i \geq 1, \quad \mathbb{E}[\|Y\varepsilon_i\|_1] \underset{n \to +\infty}{\longrightarrow} 0$$

However, by Holder inequality

$$\mathbb{E}[\|Y\varepsilon_i\|_1] \leq np(\max_k \|y_k\|_\infty) \leq \frac{1}{n^\alpha} \xrightarrow[n\to+\infty]{} 0$$

And, we can also have the result for $\|\cdot\|_2$ by the fact that $\|\cdot\|_2 \leq \|\cdot\|_1$.

Similar reasoning leads to the result of the case $p = \frac{(\max_k \|y_k\|_\infty)^{-1}}{n}$.

Last, for the case $p > \frac{(\max_k \|y_k\|_\infty)^{-1}}{n}$ and $\|g\|_\infty > \delta$, let

$$\tau = \inf\{s > 1; \ |g(x_1(s), x_2(s))| > \delta\}$$

Then, $1 < \tau < +\infty$, thanks to the regularity of $g$ and the fact that the considered differential system is autonomous. Hence,

$$
\begin{aligned}
\mathbb{E}[\sup_{t\in T} \|x(t) - x_\varepsilon(t)\|_1] &\geq \mathbb{E}\left[\int_1^\tau \sum_{i,j} \varepsilon_{i,j} |g(x_i(s), x_j(s))| \, ds\right] \\
&\geq \tau\delta \sum_{i,j} \mathbb{E}[\varepsilon_{i,j}] \\
&\geq \delta(\max_k \|y_k\|_\infty)^{-1} \\
&\geq \delta.
\end{aligned}
$$

### A.2 PROOF OF PROPOSITION 2

The proof proceeds similarly as for proposition 1, except this time the noise is not non-negative. Hence, using the properties of the absolute value and Gaussian distributions, we get

$$\mathbb{E}[\|Y\varepsilon_i\|_1] \leq \left(\sum_j |\varepsilon_{i,j}|\right)(\max_k \|y_k\|_\infty) = n\sigma(\max_k \|y_k\|_\infty) \leq \frac{1}{n^\alpha} \xrightarrow[n\to+\infty]{} 0$$

since $\mathbb{E}|\varepsilon_{i,j}| = \sigma$. The remaining cases can be shown in an analogus way.

## B  EXPERIMENTAL SET UP & ADDITIONAL NUMERICAL RESULTS

### B.1  EXPERIMENTAL SET UP

We train the competitor methods with the hyper-parameters optimized by the respective authors Shao et al. (2022) and Guo et al. (2021). However, for a fair comparison we limit the training time to the amount required by the fastest method. We report estimated training time for each method on the PEMS03 dataset in table 1, for the number of epochs suggested by the respective authors. The observed significant gain in training time notably comes from the fact that NCF is designed to perform the time evolution in a much smaller space.

### B.2  ADDITIONAL NUMERICAL RESULTS

We report additional comparison results on the datasets PEMS03 and PEMS07 proposed by Guo et al. (2021) similarly as PEMS04 and PEMS08 which we considered in section 6. Furthermore, we report error bars for the results presented in section 6. The very high error bars for ASTGNN come from the fact that under

Table 1: Training time for each of the forecasting methods

| Method | NCF | ASTGNN | D2STGNN |
|---|---|---|---|
| Time (hours) | 3.41 | 19.57 | 6.83 |

network perturbation its predictions across horizons (from 1 step ahead to 12 steps ahead) varies a lot, leading to a large standard deviation. Higher number of samples would be needed to reduce the variance. However, since previous studies (Shao et al., 2022; Liu et al., 2023) have shown that D2STGNN outperforms ASTGNN and NCF outperforms D2STGNN, we conclude that the improvement is statistically significant.

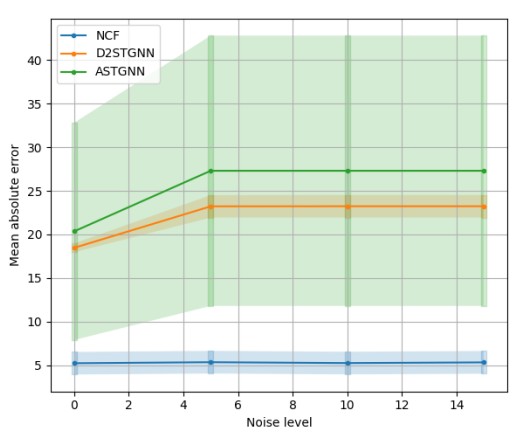

Figure 11: Test error evolution for PEMS04 dataset - 30% corrupted test data with discrete noise, $1\sigma$ error bars

Figure 12: Test error evolution for PEMS08 dataset - 30% corrupted test data with discrete noise

### B.3 ANALYSIS OF THE RESULTS

We note that the competitor methods incur a higher error after perturbing a topology, but beyond a certain level the perturbation effect becomes negligible. This can be explained by the fact that both methods extract spatial information not only from the graph but also from the time series itself. On the other hand, that might suggest the static graph information could be leveraged in a better way. Yet, given the inherent noise in real-world time series, the proposed method NCF that is designed to be robust, features much better performance.

### B.4 COMPARISON TO STATE-SPACE MODELS

We explore the performance of state-space models in the context of graph forecasting under topology perturbation. Specifically, we compare against the very recent approach mspace proposed by Rahman and Coon. We report in figures 15 and 16 comparisons on the PEMS04 and PEMS08 datasets. These datasets were chosen since mspace performs best on them according to the authors (Rahman and Coon). From our experiments, we note that NCF has an improved performance compared to mspace . We speculate that mspace is not affected by change in topology due to the fact that the state functions used by mspace do not

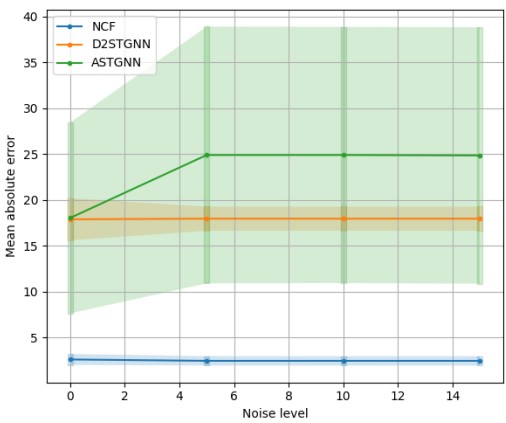

Figure 13: Test error evolution for PEMS03 dataset - 30% corrupted test data with discrete noise, $1\sigma$ error bars

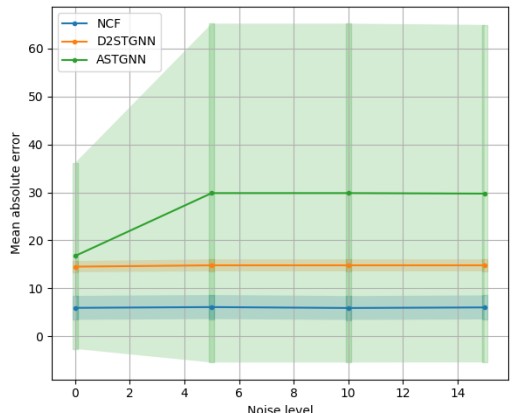

Figure 14: Test error evolution for PEMS07 dataset - 30% corrupted test data with discrete noise

depend in a strong way on the graph topology. However, that also leads to weaker performance. An ideal method should exploit the graph structure as much as possible while maintaining robustness to perturbations.

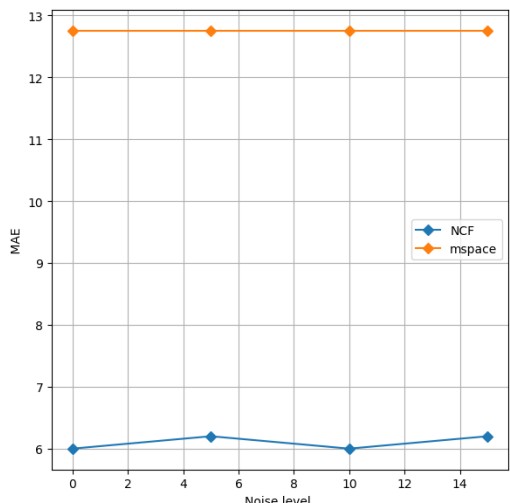

Figure 15: Test error evolution for PEMS04 dataset - 30% corrupted test data with discrete noise

Figure 16: Test error evolution for PEMS08 dataset - 30% corrupted test data with discrete noise

## B.5 COMPARISON TO ODE-BASED MODELS

We further explore the performance of NCF for higher dimensional systems and we compare it to the recent ODE-based graph forecasting approach DiskNet (Li et al., 2024b). We conduct experiments on datasets generated by Li et al. (2024b) with a Barabási–Albert topology of dimension 2000 as well as a real-world web network of dimension 4252. The dynamics are generated according to the Hindmarsh-Rose model. We observe the under topology perturbation NCF once again outperforms the competitor method. We report the mean absolute error in figures 17 and 18.

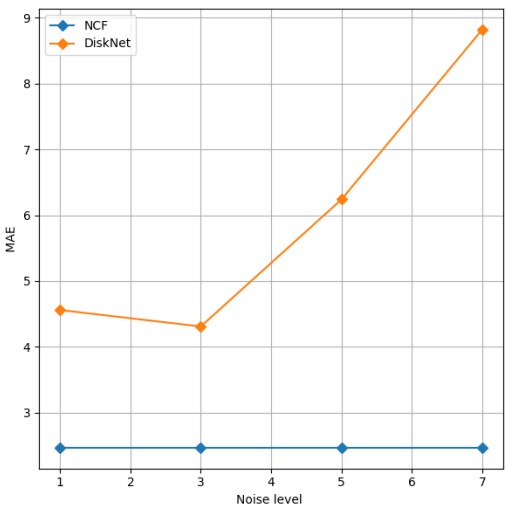 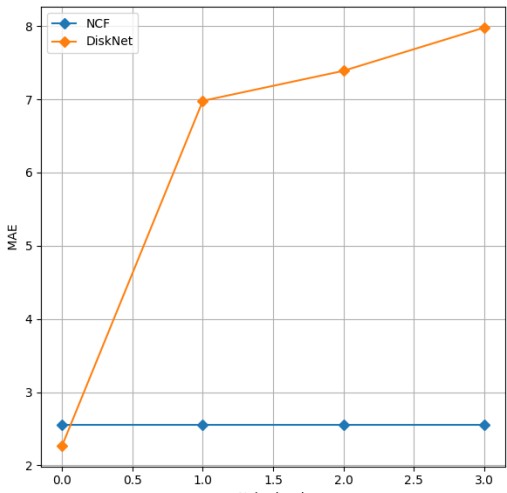

Figure 17: Test error evolution for social web dataset - 10% corrupted test data with discrete noise

Figure 18: Test error evolution for Barabási–Albert dataset - 10% corrupted test data with discrete noise

## B.6 CORRELATED NOISE

We further evaluate NCF in high-dimensional correlated noise settings on Barabási–Albert and social web networks from Li et al. (2024b). We compare against the most recent state-of-the-art approach DiskNet from Liu and Zhang (2022). We report the results in figures 19 and 20 respectively, with additional noise in the node values. We note that NCF outperforms DiskNet as soon as there is noise in the topology, hence maintaining stable performance when tested on correlated noise.

## B.7 SELECTED NODES ANALYSIS

We perform a qualitative analysis of the type of nodes that are being selected by NCF. For that matter, we compare the average values of the selected nodes across time with the global average as well as with average values for nodes with highest centrality and highest dynamic variance. We report the results in figures 21 and 22. We note that NCF selects nodes that track best the global system behavior as expected, unlike all selection methods.

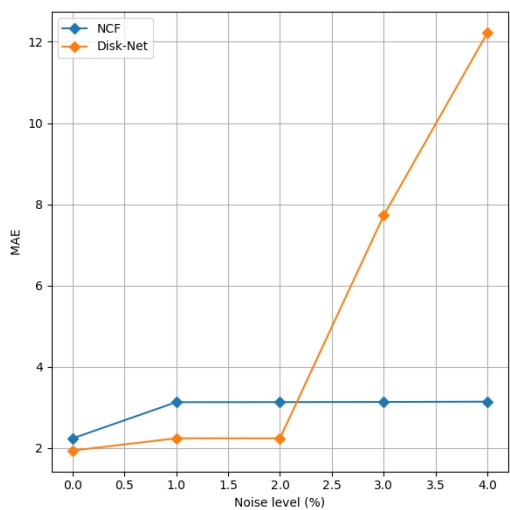

Figure 19: Test error evolution for social web dataset - 10% corrupted test data with discrete noise

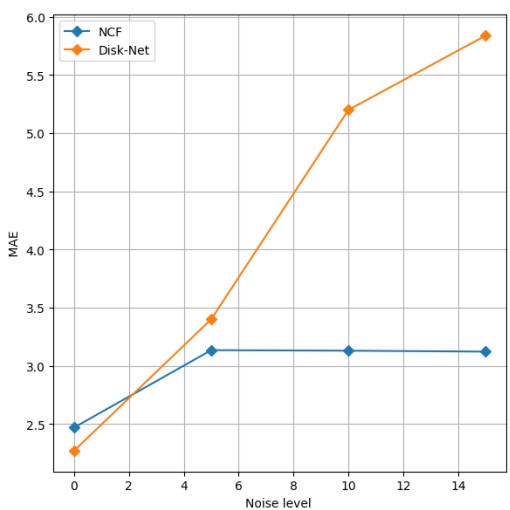

Figure 20: Test error evolution for Barabási–Albert dataset - 10% corrupted test data with discrete noise

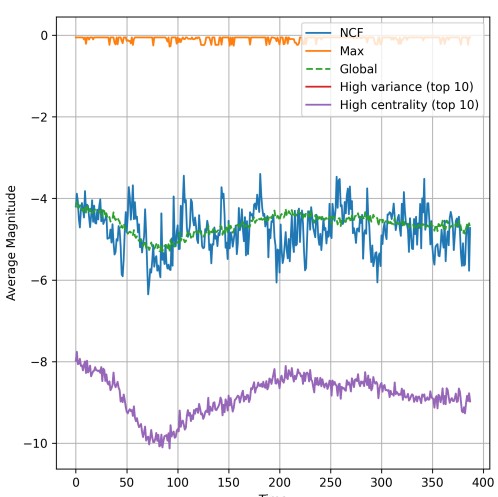

Figure 21: Average value evolution of selected nodes on the Barabási–Albert network

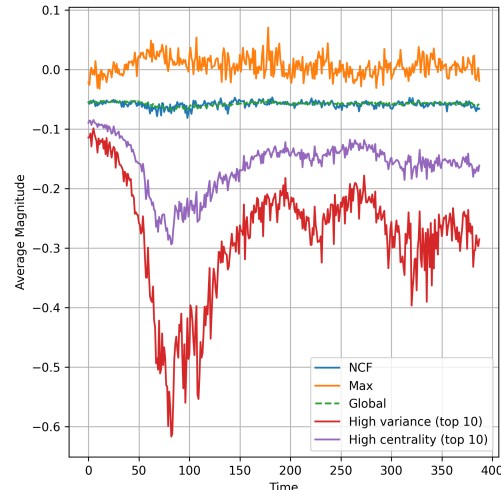

Figure 22: Average value evolution of selected nodes on the social web network

## C  NETWORK SPARSE APPROXIMATION

In the context of network misspecification, we explore the effect of sparsifying the topology on the predictability of network systems. For that matter, we estimate the closest to the ground truth sparse adjacency matrix, in terms of its predictability of the system trajectories. We leverage the fact that a given matrix $\hat{A}$ will lead to

accurate predictions of a network system -supported on a lower dimensional space of dimension $m \ll n$- if for a given set of spatial modes $y_1, \ldots, y_m$, we have $\hat{A}y_k = Ay_k$, for all $k \leq m$, as shown in Prasse and Van Mieghem (2022). More precisely, we compute $y_1, \ldots, y_m$ by Principal Component Analysis (PCA) then solve for $\hat{A}$ the following sparse optimization problem

$$\min_{\hat{A} \geq 0} \sum_{k=1}^{m} \left\| \hat{A}y_k - Ay_k \right\|_2 + \lambda \left\| \hat{A} \right\|_1 \ .$$

We report in figure 23 the prediction error of each previously tested method on the sparse topologies obtained for a subset of 2000 screen-shots of the PEMS04 traffic dataset. The results suggest that robustness to sparsity aligns with robustness to random perturbations or more generally noisy input, posing an interesting theoretical question about this relationship, which we leave for future work.

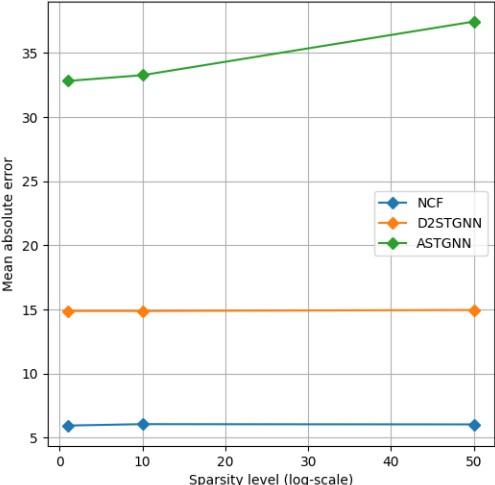

Figure 23: Test error evolution for PEMS04 dataset for varying sparsity levels

**Remark.**

We note that this simple test could serve as a benchmark for optimality of network forecasting methods or their effectiveness in capturing the fundamental characteristics and directions of highest variability of the underlying system.

## D   CODE AND PSEUDO-CODE

An implementation of the proposed method is made available at code. Additionally, a pseudo-code of the method is proposed below.

---

**Algorithm 1** Bayesian Coreset Forecasting with GNN-RNN and Encoder-Decoder Training

---

**Require:** Time series node states $\{x_1^t, \ldots, x_n^t\}_{t=1}^T$, adjacency matrix $A = (a_{ij})_{i,j=1}^n$, GNN encoder $f_{\text{GNN}}$, GNN decoder $g_{\text{GNN}}$, RNN model $f_{\text{RNN}}$, Bayesian coreset selector $\mathcal{C}$, forecasting horizon $H$
**Ensure:** Forecasted node states $\{\hat{x}_i^{T+h}\}_{i=1}^n$ for $h = 1, \ldots, H$
        ▷ — Encoder-Decoder Training with Coreset Reduction —
1:  **for** each training iteration **do**
2:     **for** $t = 1$ to $T$ **do**
3:         $H^t \leftarrow f_{\text{GNN}}(A, \{x_i^t\}_{i=1}^n)$          ▷ Encode full graph embeddings
4:         $\widetilde{H}^t \leftarrow \mathcal{C}(H^t)$          ▷ Select coreset embeddings
5:         $\hat{H}^t \leftarrow g_{\text{GNN}}(A, \widetilde{H}^t)$          ▷ Decode back to full embedding
6:     **end for**
7:     Compute reconstruction loss:

$$\mathcal{L}_{\text{rec}} = \sum_{t=1}^T \|H^t - \hat{H}^t\|^2$$

8:     Update parameters of $f_{\text{GNN}}$ and $g_{\text{GNN}}$ via backprop using $\mathcal{L}_{\text{rec}}$
9:  **end for**
        ▷ — Forecasting using trained encoder-decoder and RNN —
10:  **for** $t = 1$ to $T$ **do**
11:     $H^t \leftarrow f_{\text{GNN}}(A, \{x_i^t\}_{i=1}^n)$          ▷ Embed node features with GNN
12:     $\widetilde{H}^t \leftarrow \mathcal{C}(H^t)$          ▷ Select Bayesian coreset embeddings
13:  **end for**
14:  Initialize RNN hidden state $h^0$
15:  **for** $t = 1$ to $T$ **do**
16:     $h^t \leftarrow f_{\text{RNN}}(\widetilde{H}^t, h^{t-1})$          ▷ Evolve embeddings over time
17:  **end for**
18:  **for** $h = 1$ to $H$ **do**
19:     $h^{T+h} \leftarrow f_{\text{RNN}}(\text{NULL}, h^{T+h-1})$          ▷ Rollout without new input
20:     $\widetilde{H}^{T+h} \leftarrow \text{Decoder}(h^{T+h})$          ▷ Decode reduced embedding
21:     $\{\hat{x}_i^{T+h}\}_{i=1}^n \leftarrow g_{\text{GNN}}(A, \widetilde{H}^{T+h})$          ▷ GNN decode to full node states
22:  **end for**
23:  **return** $\{\hat{x}_i^{T+h}\}_{i=1}^n$ for $h = 1, \ldots, H$

---

