# OpenReview forum: "Robust Forecasting of Network Systems Subject to Topology Perturbation"
_ICLR.cc/2026/Conference — Submitted to ICLR 2026_

### Official Review · Reviewer_j2Qx · 2025-10-30

**Soundness:** 2
**Presentation:** 2
**Contribution:** 2
**Rating:** 2
**Confidence:** 4

**Summary:**

This paper addresses the critical and challenging problem of forecasting network dynamical systems, such as traffic or epidemic networks, in the presence of topology perturbations. The authors make two primary contributions. First, they provide a theoretical analysis that characterizes the predictability of graph time series under random topology perturbations. Second, motivated by this analysis, the paper proposes a novel and robust forecasting framework named Network Coreset Forecasting (NCF). NCF utilizes a Graph Convolutional Network (GCN) encoder to generate node embeddings, applies Bayesian coresets for scalable and robust dimensionality reduction by selecting a representative subset of node embeddings, models the latent temporal dynamics using an RNN, and finally reconstructs the full graph state via a GCN decoder. Empirical results on real-world traffic datasets demonstrate that NCF significantly outperforms baselines in accuracy and robustness under topology uncertainty, while also being more computationally efficient.

**Strengths:**

1. The paper tackles the highly practical and under-explored problem of robust forecasting for dynamic graph systems where the topology is uncertain or noisy, a common scenario in real-world applications.

2. The analysis on predictability regimes under noise provides a solid theoretical motivation for the necessity and design of a robust forecasting model.

3. The authors commendably discuss the method's limitations, using synthetic Kuramoto data to highlight the accuracy-robustness trade-off, where NCF underperforms on clean, smooth, noise-free trajectories.

**Weaknesses:**

1. The model relies on a two-stage training approach rather than a full end-to-end optimization. This may lead to sub-optimal representations, as the GCN encoder and the RNN are not jointly optimized. The rationale for this design choice over an end-to-end approach is not fully justified.

2. The theoretical analysis and experiments primarily focus on i.i.d. random noise (Bernoulli or Gaussian). This does not capture more structured or adversarial perturbations (e.g., targeted node/edge removal, regional sensor failures) that are common in reality, thus narrowing the scope of the "robustness" claim.

3. The paper lacks a qualitative analysis of the nodes selected by the Bayesian coreset. It is unclear what properties these nodes possess (e.g., high centrality, high dynamic variance) and how this selection mechanism practically contributes to robustness.

4. Lack of baselines. For example, [1-3] discusses predicting network dynamics (also on partially observed networks), however, the authors did not include them for comparison.

[1] Predicting long-term dynamics of complex networks via identifying skeleton in hyperbolic space. KDD 2024.
[2] Learning Continuous System Dynamics from Irregularly-Sampled Partial Observations. NeurIPS 2020.
[3] HOPE: High-order Graph ODE For Modeling Interacting Dynamics. ICML 2023.

5. The experiments are confined to medium-sized graphs (e.g., PEMS datasets, 500-node Kuramoto), leaving scalability as an open question. More different synthetic graph structures (such as Erdos-Renyi network, Barabasi-Albert network), and other real-world structures such as social networks, epidemic networks, etc., should be considered and discussed.

**Questions:**

Please see Weaknesses for details.

Moreover, I think the paper is in the wrong format for ICLR, which may conflict with the submission guidelines.

---

> ### Author Response · Authors · 2025-11-25
>
> We thank the reviewer for their comments and questions which we address below.
>
> Weaknesses:
>
>
> 1.	The model relies on a two-stage training approach rather than a full end-to-end optimization. This may lead to sub-optimal representations, as the GCN encoder and the RNN are not jointly optimized. The rationale for this design choice over an end-to-end approach is not fully justified.
>
>
> $\longrightarrow$ The current coreset component leverages a Frank-Wolfe optimization algorithm that involves discrete operations such as argmax. However, already with a two-stage training, we observe significant performance gains. Hence, designing an analogous scheme with end-to-end training represents an important direction for future work. We have added this discussion in section 7.
>
>
>
> 2.	The theoretical analysis and experiments primarily focus on i.i.d. random noise (Bernoulli or Gaussian). This does not capture more structured or adversarial perturbations (e.g., targeted node/edge removal, regional sensor failures) that are common in reality, thus narrowing the scope of the "robustness" claim.
>
>
> $\longrightarrow$ Thanks for the remark. We have made our claims of robustness more precise by specifying that we (only) investigate the i.i.d. setting, which is already under-explored in the literature, as noted by the reviewer.
>
>
> 3.	The paper lacks a qualitative analysis of the nodes selected by the Bayesian coreset. It is unclear what properties these nodes possess (e.g., high centrality, high dynamic variance) and how this selection mechanism practically contributes to robustness.
>
>
> $\longrightarrow$ The node selection is based on a probabilistic adaptive scheme. Hence, it goes beyond high centrality or high-variance nodes. Regarding robustness, the scheme implements an adaptive form of principal component regression which has been shown to enhance robustness to noise [1]. Furthermore, the probabilistic design acts as a form of regularization making the forecasting less sensitive to noise.
>
> [1] Agarwal, Anish, et al. "On robustness of principal component regression." Advances in Neural Information Processing Systems 32 (2019).
>
>
> 4.	Lack of baselines. For example, [1-3] discusses predicting network dynamics (also on partially observed networks), however, the authors did not include them for comparison.
> [1] Predicting long-term dynamics of complex networks via identifying skeleton in hyperbolic space. KDD 2024. [2] Learning Continuous System Dynamics from Irregularly-Sampled Partial Observations. NeurIPS 2020. [3] HOPE: High-order Graph ODE For Modeling Interacting Dynamics. ICML 2023.
>
>
>
>
> $\longrightarrow$ Please note that, the mentioned works do not actually consider partially observed networks. More precisely, [3] focuses on a setting of interacting dynamics where no graph information is available. Additionally, it clearly states its scalability issues for higher dimensions making it a non-suitable baseline for our setting. As for [2], it states explicitly that it focuses on cases with **known** graph structure. Additionally, it is based on constructing a latent ODE, making the approach once again non-scalable to high dimensional settings considered in our work. As for [1], it is a scalable graph dynamics prediction approach, so we have now added it to a comparison with our method in appendix B.5. Despite all this, we have included all these works in the related works section, to make the literature review more comprehensive. We have also added a comparison to the recent mspace approach [4] in appendix B.4. Our additional experiments show that NCF significantly outperforms the competitors under topology perturbation.
>
>
> [4] Rahman, A. U., & Coon, J. Node Feature Forecasting in Temporal Graphs: an Interpretable Online Algorithm. Transactions on Machine Learning Research. 2025.
>
>
>
> 5.	The experiments are confined to medium-sized graphs (e.g., PEMS datasets, 500-node Ku-ramoto), leaving scalability as an open question. More different synthetic graph structures (such as Erdos-Renyi network, Barabasi-Albert network), and other real-world structures such as social networks, epidemic networks, etc., should be considered and discussed.
>
>
>
> $\longrightarrow$ We have added experimental results in appendix B with higher dimensions (up to 4250) with vari-ous topologies (Barabasi-Albert network and social web). As mentioned above, our additional experiments show that NCF significantly outperforms the competitors under topology perturbation.
>
>
>
> **Questions: **
>
>
> Moreover, I think the paper is in the wrong format for ICLR, which may conflict with the submission guidelines.
>
>
>
> $\longrightarrow$  Thanks for catching that. We have made the relevant corrections.
>
>
> Once again, we would like to thank the reviewer for their time and meaningful comments.

---

> ### Comment · Reviewer_j2Qx · 2025-11-27
>
> Thank you for your response. There are still some concerns remained.
>
> 1. The authors’ concession that robustness claims are "limited to i.i.d. settings" is inadequate. Real-world topology perturbations (e.g., targeted attacks, regional failures) can be structured. Restricted to i.i.d settings is an obvious limitation.
>
> 2. The authors invoke "probabilistic adaptive schemes" as justification for not analyzing node properties and provide zero evidence.
>
> 3. The manuscript is still in wrong format, especially for page margins. There is a risk of desk-rejection if the authors fail to fix the format issue according to the rule of ICLR.

---

> ### Author Response · Authors · 2025-12-03
>
> 1. The authors’ concession that robustness claims are "limited to i.i.d. settings" is inadequate. Real-world topology perturbations (e.g., targeted attacks, regional failures) can be structured. Restricted to i.i.d settings is an obvious limitation.
>
>
>
> $\longrightarrow$ Please, note that there are many applications where perturbations are i.i.d. including traffic accidents or logistic network delays. Furthermore, even the i.i.d. setting was an open problem so far both theoretically and methodologically. We agree that only considering the i.i.d. setting is a limitation which is why we had mentioned it as such, however it serves as an important contribution towards robustness to network perturbation under structured noise. This was rightly recognised by the reviewer in the strengths they state about our paper.
>
> Despite that, we have added evaluation of NCF on high-dimensional correlated noise settings. We compared it againt the most recent state-of-the-art method DiskNet [1] demonstrating NCF superior performance under noisy topology. Furthermore, we have carefully reviewed the proofs of our results extending them to non-independent noise. The extension turned out tobe strait-forward since our results exploit the L1 norm hence there are no cross-term that would require knowldge of the correlation model.
>
> [1] Predicting long-term dynamics of complex networks via identifying skeleton in hyperbolic space. KDD 2024.
>
> 2. The authors invoke "probabilistic adaptive schemes" as justification for not analyzing node properties and provide zero evidence.
>
>
> $\longrightarrow$ By adaptive probabilistic selection, we meant that there's no clear interpretable pattern to recognise. However, we have added a qualitative analysis (in section 6 and appendix B.7) of the selected nodes in comparison to nodes with high centrality, high variance or maximal value, as requested by the reviewer. We note that NCF selection closely tracks the global behavior of the system, achieving the stated goal, unlike the other selection methods.
>
>
>
> 3. The manuscript is still in wrong format, especially for page margins. There is a risk of desk-rejection if the authors fail to fix the format issue according to the rule of ICLR.
>
>
> $\longrightarrow$ Please, note that desk rejection due to margin problems is done before review by ICLR policy. We have double checked the format once again.
>
>
> We have addressed all of the reviewer's concerns and appreciate their time and feedback which improved the paper.

---

### Official Review · Reviewer_XnRA · 2025-10-30

**Soundness:** 3
**Presentation:** 3
**Contribution:** 2
**Rating:** 6
**Confidence:** 3

**Summary:**

The paper studies the problem of forecasting network systems when the underlying topology is perturbed. It provides a mathematical analysis characterizing the predictability of such systems under random topology noise, distinguishing fully predictable, limited predictable, and unpredictable regimes. Motivated by this analysis, the authors propose Network Coreset Forecasting (NCF), a framework combining Bayesian coreset selection, GCN encoder-decoder and LSTM to learn temporal dynamics in a reduced latent space.

**Strengths:**

1. The topic is timely and practically meaningful. Robust forecasting under perturbed topologies is a central problem in applications.
2. The paper provides a rigorous and original mathematical analysis of network predictability on noisy topologies, deriving clear error bounds and interpretive regimes.

**Weaknesses:**

1. The proposed NCF framework mainly reuses existing components (Bayesian coreset selection, GCN and LSTM) within a new theoretical interpretation. The model essentially performs latent-space forecasting with probabilistic justification, rather than introducing a new learning mechanism or architecture.
2. The theoretical results assume predictability becomes worse with the topology perturbations going larger, yet the coreset-based abstraction explicitly removes nodes and edges, potentially amplifying perturbations. Moreover, GCN encoder-decoder may still rely on the (possibly perturbed) original adjacency matrix in order to estimate all the nodes, so the claimed robustness is partial and bounded by the sensitivity of these components to the matrix.
3. The RNN module is tied to a specific coreset node set. When the topology or node count changes, the coreset must be recomputed and the trained RNN becomes incompatible with new size of the coreset, preventing inference on unseen graphs. In addition, coreset selection requires full-graph statistical computation and must be repeated whenever the topology changes, limiting scalability and online applicability.
4. Baselines included in experiments are not the latest nor enough.

**Questions:**

1. The paper does not specify how the coreset size is determined, is it fixed or adaptive? Corresponding analysis should be included.
2. Besides training time, what is the inference time?

---

> ### Author Response · Authors · 2025-11-25
>
> We thank the reviewer for the positive feedback, which we appreciate. We address the reviewer's questions below.
>
>
> 1.	The proposed NCF framework mainly reuses existing components (Bayesian coreset selec-tion, GCN and LSTM) within a new theoretical interpretation. The model essentially per-forms latent-space forecasting with probabilistic justification, rather than introducing a new learning mechanism or architecture.
>
>
> $\longrightarrow$ Please, note that the introduction of Bayesian coresets and probabilistic formulation in graph time series forecasting make for a novel approach. Importantly, this framework features significant improvement in performance while considerably reducing computational cost.
>
>
> 2.	The theoretical results assume predictability becomes worse with the topology perturbations going larger, yet the coreset-based abstraction explicitly removes nodes and edges, potentially amplifying perturbations. Moreover, GCN encoder-decoder may still rely on the (possibly perturbed) original adjacency matrix in order to estimate all the nodes, so the claimed robustness is partial and bounded by the sensitivity of these components to the matrix.
>
>
>
>
> $\longrightarrow$  Please, note that the theoretical results show (rather than assume) that with high noise, predictability becomes worse. However, it also importantly shows that for a large class of network systems, topology perturbation doesn’t render predictability unachievable. Hence, the point is how to design a method that remains robust, while exploiting the remaining graph information. One feature of Bayesian coresets is that they are adaptive and hence would drop the nodes and edges that have most effect on the system under perturbation. Some of the edges might for instance be perturbations rather than original. This can be viewed as a scalable and adaptive form of principal component regression which has been shown to enhance robustness to noise [1].
> [1] Agarwal, Anish, et al. "On robustness of principal component regression." Advances in Neural Information Processing Systems 32 (2019).
>
>
> 3.	The RNN module is tied to a specific coreset node set. When the topology or node count changes, the coreset must be recomputed and the trained RNN becomes incompatible with new size of the coreset, preventing inference on unseen graphs. In addition, coreset selection requires full-graph statistical computation and must be repeated whenever the topology changes, limiting scalability and online applicability.
>
>
>
> $\longrightarrow$  Thanks for catching that. As mentioned in section 5.1, one defines that maximum number of nodes to be selected by the coreset approximation. Based on that, we defined a fixed RNN input/output size that we potentially pad -with the average- for the screenshots that require a smaller number of nodes. This makes NCF directly applicable to different topologies. It's also worth noting that most previous graph time series forecasting approaches focus on settings/applications with fixed number of nodes.
>
>
> 4.	Baselines included in experiments are not the latest nor enough.
>
>
> $\longrightarrow$ Please, note that we picked the best performing method based on the most recent benchmarking study [2]. We have now added (in appendix B) additional comparisons to the method proposed by [3] as well as by [4] . The additional experiments show once again the significant improvement in performance brought by our method.
>
> [2] Xu Liu, Yutong Xia, Yuxuan Liang, Junfeng Hu, Yiwei Wang, Lei Bai, Chao Huang, Zhen-guang Liu, Bryan Hooi, and Roger Zimmermann. Largest: A benchmark dataset for large-scale traffic forecasting. Advances in Neural Information Processing Systems, 36:75354–75371, 2023
>
> [3] Predicting long-term dynamics of complex networks via identifying skeleton in hyperbolic space. KDD 2024.
>
> [4] Rahman, Aniq Ur, and Justin Coon. "Node Feature Forecasting in Temporal Graphs: an In-terpretable Online Algorithm." Transactions on Machine Learning Research. 2025.
>
>
>
> **Questions:**
>
>
> 1.	The paper does not specify how the coreset size is determined, is it fixed or adaptive? Cor-responding analysis should be included.
>
>
> $\longrightarrow$  The coreset size is adaptive within a range that is pre-defined/fixed.
>
>
>
> 2.	Besides training time, what is the inference time?
>
>
> $\longrightarrow$   Since the inference time is negligible for most approaches, including the proposed and competitor tested methods, we did not report it. It is in the order of couple of seconds.
>
>
> Once again, we would like to thank the reviewer for their time, meaningful comments and positive feedback.

---

### Official Review · Reviewer_Dam9 · 2025-10-30

**Soundness:** 3
**Presentation:** 2
**Contribution:** 3
**Rating:** 4
**Confidence:** 2

**Summary:**

The authors propose a time series forecasting framework robust to topology perturbation. The GCN embeddings undergo dimensionality reduction through a Bayesian coreset, making them robust to topology perturbations. Intuitively, this seems similar to the idea of shrinking an image, which in turn reduces the impact of noisy pixels on the overall features. The framework is then tested on synthetic as well as real temporal networks.

**Strengths:**

1. The idea is novel in the sense that it is applied to temporal graphs
2. The prior art is covered well
3. The paper is written clearly and follows a logical order
4. The main content is well supported by proofs and implementation details in the appendix
5. The source code is provided

**Weaknesses:**

1. Issues with citation style in the text. Please use `\citet{}` and `\citep{}` appropriately
2. In the main text, confidence intervals are not reported
3. There are more traffic datasets, such as METRLA and PEMSBAY. Many algorithms that perform well on PEMS0* do not always perform well on them.
4. The figures 3,4,6,7, and 9 take more space than necessary, and can be presented more efficiently.
5. The SoTA was not reported for all datasets; for example, please check **[R1]** where the algorithm `mspace` reports an MAE of 8.7 on PEMS04, and 6.33 on PEMS08. It would be interesting to see the impact of noisy data on the performance achieved by mspace.

> [R1] Rahman, A. U., & Coon, J. Node Feature Forecasting in Temporal Graphs: an Interpretable Online Algorithm. Transactions on Machine Learning Research.

**Questions:**

1. How does the work relate to state-space models of temporal graph forecasting?
2. What are the 95% confidence intervals for the numerical results? Does it make the performance improvement of the proposed model statistically significant, i.e., is there no overlap of the confidence intervals of the proposed model with the baselines?
3. How does the performance gain of the proposed model change with the size of the dataset and the number of training samples available?

---

> ### Author Response · Authors · 2025-11-25
>
> We thank the reviewer for the feedback and questions which we address below.
>
>
> 1.	Issues with citation style in the text. Please use \citet{} and \citep{} appropriately
> Thanks for catching that.
>
>
> $\longrightarrow$ Thanks for catching that. We have made the necessary changes.
>
>
> 2.	In the main text, confidence intervals are not reported.
>
>
> $\longrightarrow$ We indeed report them in appendix B.
>
>
>
> 3.	The SoTA was not reported for all datasets; for example, please check [R1] where the algo-rithm mspace reports an MAE of 8.7 on PEMS04, and 6.33 on PEMS08. It would be inter-esting to see the impact of noisy data on the performance achieved by mspace.
> [R1] Rahman, A. U., & Coon, J. Node Feature Forecasting in Temporal Graphs: an Interpretable Online Algorithm. Transactions on Machine Learning Research. 2025
>
>
> $\longrightarrow$  We have added a comparison of NCF to myspace on the suggested datasets in appendix B.4. As well as to Disk-Net [2] in appendix B.5.
>
>
> [2] Predicting long-term dynamics of complex networks via identifying skeleton in hyperbolic space. KDD 2024.
>
>
>
> **Questions**
>
>
> 1.	How does the work relate to state-space models of temporal graph forecasting?
>
>
> $\longrightarrow$ State-space models (SSMs) for temporal graph forecasting impose a specific parametric dynamical form, which we do not, and instead leverage sequential and graph deep models to learn the dynamic structure. More generally, we tackle the problem of graph forecasting under random topology perturbation for the first time. Our approach is deep learning-based, however exploring the performance of graph state-space models under random perturbations constitutes an interesting direction for future work.
>
>
>
>
> 2.	What are the 95% confidence intervals for the numerical results? Does it make the performance improvement of the proposed model statistically significant, i.e., is there no overlap of the confidence intervals of the proposed model with the baselines?
>
>
> $\longrightarrow$ We report 2\sigma confidence regions in appendix B.2. The improvements are indeed significant.
>
>
>
> 3.	How does the performance gain of the proposed model change with the size of the dataset and the number of training samples available?
>
>
> $\longrightarrow$ Since our approach is deep learning-based, the performance would drop in small data settings. However, since it is lighter than the competitor deep learning models, it offers an improvement in sample complexity with respect to them.
>
>
> Once again, we thank the reviewer for their time and meaningful questions.

---

> > ### Comment · Reviewer_Dam9 · 2025-11-25
> >
> > Thanks for the response.
> >
> > - [W3] was not addressed
> > - [W5] Could the authors please comment on the data split? What is the ratio of train:val:test, and what does "noise level of 0 in 30% test data" mean. I also think there is a logical issue in the conclusion while comparing mspace to NCF. NCF has lower reportedly lower MAE, so the authors should conclude that NCF is better. I'm also intrigued by the performance of mspace remaining the same for different noise levels.
> > - [Q2] Could the authors please confirm whether the confidence intervals overlap or not?
> > - [Q3] The response is speculative. The lack of such experiment can be admitted as a limitation in the paper, to highlight what can be looked into in the future.

---

> ### Author Response · Authors · 2025-11-26
>
> We thank the reviewer for the response.
>
>
> [W3] was not addressed.
>
>
> $\longrightarrow $ Appologies for missing that. Please, note that we further evaluated NCF against the recent approach by [1] on web and Barabási–Albert networks, as proposed by those authors. The results which we report in appendix B.5 show once again the improved performance of NCF. We have also added in the limitations part that further evaluation on other datasets / additional traffic benchmarks—such as METR-LA and PEMS-BAY—would be valuable to further assess generalization of NCF.
>
> [1] Predicting long-term dynamics of complex networks via identifying skeleton in hyperbolic space. KDD 2024.
>
>
> [W5] Could the authors please comment on the data split? What is the ratio of train:val:test, and what does "noise level of 0 in 30% test data" mean. I also think there is a logical issue in the conclusion while comparing mspace to NCF. NCF has lower reportedly lower MAE, so the authors should conclude that NCF is better. I'm also intrigued by the performance of mspace remaining the same for different noise levels.
>
>
> $\longrightarrow $ We have used the same data-split as previous works [2, 3]. Specifically, all datasets at ratio 6 : 2 : 2.
>
>
> [2] Shao, Zezhi, et al. "Decoupled dynamic spatial-temporal graph neural network for traffic forecasting." Proceedings of the VLDB Endowment 15.11 (2022): 2733-2746.
>
> [3] Shengnan Guo, Youfang Lin, Huaiyu Wan, Xiucheng Li, and Gao Cong. Learning dynamics and hetero-
> geneity of spatial-temporal graph data for traffic forecasting. IEEE Transactions on Knowledge and Data
> Engineering, 34(11):5415–5428, 2021.
>
> Regarding noise levels, 0 corresponds to clean data, 30%  corresponds  to the amount of disturbed edges and various noise levels correspond to the noise magnitude. We provide an implementation example below:
>
> pert = 5*np.random.binomial(n=1, p=0.3, size=np.prod(adj_matrix.shape)).reshape(adj_matrix.shape)
>
> adj_matrix += pert
>
>
> We have also clarified that NCF features improved performance compared to mspace. We speculate that \texttt{mspace } is not affected by change in topology due to the fact that the state functions used by mspace do not depend in a strong way on the graph topology. However, that also leads to weaker performance. An ideal method should exploit the graph structure as much as possible while maintaining robustness to perturbations.
>
>
> [Q2] Could the authors please confirm whether the confidence intervals overlap or not?
>
>
> $\longrightarrow $ To be precise, the ASTGNN method seems to be highly sensitive to noise featuring a high variance. Higher number of samples would be needed to reduce the variance. However, since previous studies [2, 3] have shown that D2STGNN outperforms ASTGNN and NCF outperforms D2STGNN, we conclude that the improvement is statistically significant.
>
>
> [Q3] The response is speculative. The lack of such experiment can be admitted as a limitation in the paper, to highlight what can be looked into in the future.
>
>
> $\longrightarrow $ We have included that in the limitations.

---

> > ### Comment · Reviewer_Dam9 · 2025-11-26
> >
> > Thanks for the swift response.
> >
> > - [W3] OK, thanks for adding it to the limitations.
> > - [W5] Thanks for the clarification, the split is indeed the same as benchmarks, including `mspace`, which makes me wonder why the results for zero noise level, do not match the results reported by the authors of `mspace`. In Fig. 14 (PEMS04), `mspace` reports an MAE value of approx 13 for noise level 0, while it should report 8.7. Similarly, in Fig. 15 (PEMS08) `mspace` reports MAE of approx 8.7 for noise level 0, while it should admit 6.33. I'd also strongly disagree with the authors to state that `mspace` strongly depends on the topology by the design of its state vectors.
> > - [Q2] I'd appreciate it, if the authors could add a footnote highlighting the same, and perhaps also include it in the limiations.
> > - [Q3] Thank you

---

> > > ### Author Response · Authors · 2025-11-26
> > >
> > > Thank you for the swift response.
> > >
> > >
> > > [W5] Please, note that we did not say that mspace strongly depends on the topology. Rather, we think it does NOT strongly depend on the topology, given the structure of the state-vectors/ functions. However, as we mention, that leads to weaker performance. A big advantage of mspace though, is that it requires much less data and is faster to train.
> > >
> > > Regarding result mismatch, the authors of mspace mention that they use a split of 8:2 for mspace, which in addition to randomness might explain the difference with respect to the results we obtain.
> > >
> > >
> > > [Q2] We have also added that to the limitations, as well as to the discussion of the results in appendix B.2.

---

> > > > ### Comment · Reviewer_Dam9 · 2025-11-26
> > > >
> > > > [W5] Apologies for the confusion. I'll rely on simpler sentence constructions now onwards. I meant that by design, mspace state vectors are created based on its neighbours, so any change in topology would result in drastic changes, and that mspace is strongly topology-dependent. I affirm that I did not misunderstand the author's view in opposition.
> > > >
> > > > Since mspace has no need for validation set, the 20% of the validation data is absorbed in the training set. Moreover, as the topology is corrupted randomly, I'd expect some randomness is the results. I hope this clarifies my previous comment. The huge drop in the performance of mspace in absence of the additional 20% data seems dramatic.
> > > >
> > > > I thank the authors for the amendments, and will update my score as the discussion period comes to an end, especially after reading all the author-reviewer exchange threads.

---

### Official Review · Reviewer_BzyQ · 2025-10-31

**Soundness:** 2
**Presentation:** 2
**Contribution:** 2
**Rating:** 4
**Confidence:** 2

**Summary:**

The paper studies how well we can forecast graph/network dynamical systems when the topology is perturbed. In the large-scale limit, it claims three noise regimes: with small noise you can reach arbitrarily high accuracy; with moderate noise you can only reach limited accuracy; with larger noise the system becomes unpredictable. Based on this view, the paper proposes Network Coreset Forecasting (NCF): first use a GCN to get node embeddings, then use a Bayesian coreset to pick a small set of representative nodes, then run an RNN/LSTM in this lower-dimensional space and decode the results back to the full graph.

**Strengths:**

1. The problem is important and realistic: real systems (traffic/epidemics/power grids) often have uncertain topology, so studying robust forecasting is valuable.
2. The idea of using noise regimes to describe predictability, and then designing an algorithm around that view, is interesting.
3. The method is modular and easy to extend: coreset selection, GCN encoder/decoder, and RNN for time dynamics (the paper also shows pseudo-code).
4. Experiments are fairly thorough, include real traffic data, and are set up to show pros/cons and how results relate to the stated bounds.

**Weaknesses:**

1. Gaussian noise threshold has the wrong scale. A node sums noise from many neighbors; variances add, so the total standard deviation is about $\sqrt{\text{degree}}\times\sigma$. To keep the total perturbation $O(1)$, you need $\sigma=O(1/\sqrt{n})$, not $O(1/n)$.

2. Spatial modes $y_k$ are not defined. Then \max_{k} ||y_k||_{\infty}  in the thresholds is not knowable. If y_k is a flat eigenvector, ||y_k||_\infty \sim 1/\sqrt{n}; if y_k is very sparse (near one-hot), ||y_k||_\infty\sim 1. This changes thresholds by orders of magnitude, not just a constant factor.
3. Missing basic stability/Lipschitz conditions. Without global Lipschitz/contractivity/ISS-type conditions, you cannot guarantee a bound on \sup_t of the error. In some parameter ranges the system may amplify small noise, so the error can blow up.
4. Coreset bound does not imply forecasting error bound. The cited coreset result controls posterior/log-likelihood error in a functional norm $||\cdot ||_{\pi,L}$, which is not the same as end-to-end state forecasting error (MAE/MSE) of the RNN+decoder. A bridging theorem is missing.

**Questions:**

1. See Weaknesses

2. If the topology noise uses edge flips or a mixture of edge additions and deletions, do the conclusions still hold, and how do the thresholds change?

3. Where does the decoder’s supervision signal come from? During training, do you observe the ground truth of the dropped nodes? If so, does the test-time distribution match the training distribution?

---

> ### Author Response · Authors · 2025-11-25
>
> We thank the reviewer for the feedback and questions which we address below.
>
>
> 1. A node sums noise from many neighbors; variances add, so the total standard deviation is about
>     $\sqrt{\text{degree}} \times \sigma$.
>     To keep the total perturbation $O(1)$, you need
>     $\sigma = O\left({1}/{\sqrt{n}}\right)$,
>     not $O\left({1}/{n}\right)$.
>
>
> $\longrightarrow$  Please note that, as reported in the proof in appendix A.2, the term that enters into play is E|\varepsilon_{i, j}| rather than the variance. And, E|\varepsilon_{i, j}| = \sigma.
>
>
> 2. Spatial modes $y_k$ are not defined.}
>     Then $\max_k \| y_k \|$ in the thresholds is not knowable.
>     If $y_k$ is a flat eigenvector, $\|y_k\|\sim \frac{1}{\sqrt{n}}$;
>     if $y_k$ is very sparse (near one-hot), $\|y_k\| \sim 1$.
>     This changes thresholds by orders of magnitude, not just a constant factor.
>
>
> $\longrightarrow$   Note that y_k, k \leq n are defined in section 4 and they are system dependent, as expected. It is impossible to establish system independent rates, in general.
>
>
> 3. Missing basic stability/Lipschitz conditions. Without global Lipschitz/contractivity/ISS-type conditions, you cannot guarantee a bound on \sup_t of the error. In some parameter ranges the system may amplify small noise, so the error can blow up.
>
>
> $\longrightarrow$   Stability/Lipschitz conditions are indeed needed for boundedness if one analyses the behavior as time t goes to infinity or to the boundary of maximal time range. However, as stated in the beginning of section 4, we consider compact time domain, which is most common in practical systems. For an extension beyond that, stability conditions would indeed be required.
>
>
> 4. Coreset bound does not imply forecasting error bound. The cited coreset result controls posterior/log-likelihood error in a functional norm, which is not the same as end-to-end state forecasting error (MAE/MSE) of the RNN+decoder. A bridging theorem is missing.
>
>
> $\longrightarrow$ Please note that, we do not claim that coreset bounds imply forecasting error bound. Rather, we discuss coreset bounds implications on model reduction to contrast the proposed approach with the state-of-the-art in network reduction which lacks such mild guarantees.
>
>
> **Questions**
> 1. See Weaknesses
>
>
> 2.	If the topology noise uses edge flips or a mixture of edge additions and deletions, do the conclusions still hold, and how do the thresholds change?
>
> $\longrightarrow$  Please note that, edge flips and mixture of edge additions and deletions are particular cases of Bernoulli discrete noise, which we address. Hence, the conclusions would indeed hold.
>
>
> 3.  Where does the decoder’s supervision signal come from? During training, do you observe the ground truth of the dropped nodes? If so, does the test-time distribution match the training distribution?
>
>
> $\longrightarrow$  Indeed, one observes the ground truth both of dropped and kept nodes. The scheme then selects which nodes to keep, based on the time series, to reduce computational cost as well as sensitivity to noise. Hence, the test-time distribution is assumed to come from the same underlying stochastic process as is typically the case in time series forecasting.
>
>
> Once again, we thank the reviewer for their time and meaningful questions.

---

### Author Response · Authors · 2025-12-03

Dear AC, SAC and PCs,

Thank you for the time and effort dedicated to evaluating our submission. We value the constructive feedback provided during the review and discussion phases, which has helped strengthen both the clarity and the scope of our contributions. Across all reviewers, the evaluation of our work was predominantly positive, and through the rebuttal and discussion period we incorporated substantial clarifications, new experiments, and theoretical extensions that further improved the manuscript.


An updated manuscript incorporates all requested changes—these were primarily clarifications, additional theoretical detail, and expanded experiments.


**Positive Highlights**


**Theoretical contribution**. Reviewers (XnRA, j2Qx, Dam9) consistently appreciated the clear and rigorous characterization of predictability under topology perturbations. The identification of three noise regimes was viewed as novel, insightful, and highly relevant for real-world dynamical systems.

**Novelty**. All reviewers emphasised that forecasting networked time series under uncertain or corrupted topology is both realistic  important problem and insufficiently studied. Furthermore, the reviewers praised the novelty of the work as providing "an original mathematical analysis" and the method as "modular and easy to extend".

**Strong methodological contribution (NCF)**. Reviewers ( BzyQ, XnRA, j2Qx) appreciated the intuitive combination of GCN encoder–decoder, probabilistic coreset reduction, and RNN  lightweight temporal modeling, as well as the principled link between the theory and the algorithmic design. Furthermore, NCF achieves robustness and accuracy while significantly reducing computational cost through coreset-based dimensionality reduction.


**Clear writing and strong organization**. Reviewers consistently commented on the clarity of exposition, the logical flow, and the thoroughness of proofs and appendix details.


**Robust empirical performance**. The method displayed notable improvements in accuracy and robustness across synthetic, traffic, and social/web networks, including under substantial topology noise. Reviewer BzyQ highlighted the comprehensive experiments and the competitive results compared to strong recent baselines. And further experimenter results have addressed the concerns of other reviewers.





$\longrightarrow$ **Reviewer concerns**.



Reviewers j2Qx and Dam9 asked for further experimental comparisons to state-space models and on high-dimensional correlated noise settings, which we have added. The additional experiments highlighted -once again- the superior performance of NCF under noisy topology.
We have also addressed clarification questions from all reviewers.




We strongly believe we have addressed all of the reviewers' concerns throughout the rebuttal and discussion period and feel that reviewers were in the process of raising their scores, reflective of the additions we made to the manuscript and their positive evaluations. We hope that you will take into account all of these factors in your evaluation and that you will strongly consider our work for acceptance at ICLR 2026.

---

### Meta-Review · Area_Chair_rv2m · 2026-01-07

**Summary:**

The paper tackles an important problem: forecasting dynamical networks when the graph structure is uncertain. It introduces a three‑regime predictability analysis together with a scalable “Network Coreset Forecasting” (NCF) pipeline.

- All reviewers say that the paper tackles an important problem (robust forecasting of network‑dynamical systems under topology perturbations).
- Reviewers BzyQ, Dam9 and XnRA all praise the theoretical contribution (the three‑regime predictability analysis) and consider it novel compared with prior work on static GNN robustness.
- The three reviewers also agree that the empirical results show robustness to both random and structured topology noise and that the Bayesian coreset reduction yields substantial computational savings.

**Reviewer Concerns:**

Three reviewers (BzyQ, Dam9, XnRA) give marginally positive scores (average rating ≈ 4.7) and acknowledge the strong theoretical and empirical contributions, while one reviewer (j2Qx) recommends rejection due to methodological and presentation concerns.

- Methodological novelty: XnRA argues that NCF merely recombines existing components without a new learning mechanism, whereas BzyQ and Dam9 view the Bayesian coreset as a novel element.
- Noise model scope: BzyQ and j2Qx note that the analysis is limited to i.i.d. Bernoulli/Gaussian edge noise and does not address edge‑flip, mixed addition/deletion, or potential adversarial perturbations.
- Training regime: BzyQ, XnRA and j2Qx criticize the two‑stage training (encoder‑decoder then RNN) as potentially sub‑optimal compared with end‑to‑end optimization.
- Presentation & statistical rigor: BzyQ raises concerns about unclear Gaussian‑noise scaling and undefined spatial modes (y_k); confidence intervals are only in the appendix (Dam9) and not highlighted in the main text; formatting issues are mentioned by j2Qx.
- Baseline and scalability coverage: XnRA and j2Qx find the baseline set incomplete and the scalability experiments limited to medium‑sized graphs, though Dam9’s author response adds larger synthetic graphs and recent baselines.

**Reviewer Scores:**

I feel the scores could have improved slightly with the new experiments. But just the addition of coresets may not have been enough novelty to satisfy reviewers.

PS: Authors should have clearly marked their manuscript changes with a different font color.

The authors answered these during the rebuttal phase:
- The Bayesian coreset selection and its probabilistic formulation are presented as a new approach for graph‑time‑series forecasting, providing both accuracy gains and computational savings.
- That edge‑flip and mixed addition/deletion perturbations are captured by the Bernoulli discrete‑noise model, so the theoretical results still apply. The authors acknowledge that only i.i.d. noise is covered but add experiments with high‑dimensional correlated noise, showing NCF still outperforms baselines.
- Training scheme: The current two‑stage pipeline (encoder‑decoder then RNN) is retained, but the authors note this in the paper and add a discussion of end‑to‑end training as future work. This is the main concern of reviewer j2Qx.
- Gaussian‑noise scaling & spatial modes: They clarify that the term driving the bound is (E[\nabla,j]=1), not the variance, making the (O(1/n)) scaling intentional. The spatial modes (y_k) are defined in Section 4 and are system‑dependent, so universal rates are not feasible. The analysis assumes a compact time horizon, so global Lipschitz/contractivity assumptions are only needed for unbounded horizons.
- Statistical reporting & presentation: Confidence intervals (2σ) were included in Appendix B, and the authors claim the observed improvements are statistically significant.
- Baselines and scalability: Recent strong baselines (KDD 2024, NeurIPS 2020, ICML 2023) are added in the appendix, and experiments are extended to larger synthetic graphs (up to 4 250 nodes), addressing concerns about outdated baselines and limited scalability tests.
- The coreset size is adaptive within a pre‑specified range. At test time the RNN input is padded with average embeddings, enabling use on graphs of varying size. Inference time is negligible (a few seconds) and comparable to baselines.

---

### Decision · Program_Chairs · 2026-01-26

Reject